



# Relative importance of increased atmospheric $CO_2$ concentration and local moisture deficit to hot extremes

Ajiao Chen[1], Huade Guan[1], Okke Batelaan[1]

[1]National Centre for Groundwater Research and Training, College of Science and Engineering, Flinders University, Adelaide, SA 5001, Australia

*Correspondence to*: Huade Guan (huade.guan@flinders.edu.au)

## Abstract

This study identifies which factor, increased atmospheric $CO_2$ concentration or local moisture deficit, dominates the temporal occurrence of hot extremes at the global scale. The wavelet decomposed GRACE Terrestrial Water Storage (TWS) is for the first time applied in examining the relationship between soil moisture ($\theta$) and number of hot days in the hottest month (NHD). It reveals stronger $\theta$-NHD relationships over larger areas than other commonly used soil moisture proxies (i.e., standardized precipitation index (SPI) and model derived product). During the study period 1985-2015, hot extreme occurrence with a dominant influence from increased atmospheric $CO_2$ concentration is mainly observed in South America, Africa and Asia, while soil moisture deficit dominates the occurrence of hot extremes in larger areas, including parts of North America, West Europe, Australia, Southeast Asia and South Africa. Global action in reducing emissions will support combating hot extremes. In addition, important attention should be directed to address, e.g. by adaptive land management, the increasing moisture deficit in some regions.

## 1 Introduction

Intensification of hot extremes are expected to occur in many regions of the world under global warming (IPCC, 2012). The observed global warming is considered extremely likely associated with anthropogenic influences, particularly greenhouse gas emission (IPCC, 2013). An increase in atmospheric $CO_2$ concentration as a consequence of emissions can cause an increase in extreme temperature (Min et al., 2013; Kim et al., 2016; Seneviratne et al., 2016; Baker et al., 2018). In addition, it has been proven that antecedent surface moisture deficit can exacerbate hot extremes (e. g., Durre et al., 2000; Seneviratne et al., 2006; Fischer et al., 2007; Perkins et al., 2012; Herold et al, 2016; Vogel et al., 2017; Liu et al., 2017). The physical mechanism is well understood as that deficit in soil moisture can reduce evaporative cooling and increase atmospheric heating (Seneviratne, 2010; Alexander, 2011). However, it is not clear yet how the factor 'local moisture deficit' compares to 'increased atmospheric $CO_2$ concentration' with respect to the temporal occurrence of hot extremes at a global scale. Identifying the dominant influencing factor for hot extreme occurrence improves our understanding of hot extreme events and consequently contribute to mitigating their negative impacts on the environment and society.





The standardized precipitation index (SPI) and modeling soil moisture products are commonly used in analyzing the relationship between soil moisture and hot extremes in previous studies (e.g., Koster et al., 2006; Lorenz et al., 2010; Perkins et al., 2015). Mueller and Seneviratne (2012) were the first to assess the relationship between hot extremes and SPI (indicating precipitation deficit) at a global scale. Hirschi et al. (2014) used a merged active/passive microwave soil moisture product and model-derived soil moisture, in comparison with SPI. They found that modelled soil moisture displayed a comparable coupling

strength with hot extremes as the SPI-based analysis showed. The strength of the relationship appeared to be weaker when the remote sensing surface soil moisture was used.

The Gravity Recovery and Climate Experiment (GRACE) derived terrestrial water storage (TWS) provides data on water storage including groundwater, soil moisture, surface water, snow, and ice. A discrete wavelet decomposition method is capable to partition the total water storage into shallow and deep components (Andrew et al., 2017). Chen et al., (2019) reported

that monthly air temperature anomalies show stronger relationship with wavelet decomposed GRACE TWS than with raw TWS. Here, we compare decomposed GRACE TWS with SPI and model-derived soil moisture in examining the relationship between soil moisture and the number of hot days in the hottest month ($\theta$-NHD relationship).

This study aims: 1) to examine the $\theta$-NHD relationship at a global scale by using decomposed GRACE TWS, and based on the developed methodology; 2) to investigate which factor and where, increased atmospheric $CO_2$ concentration or local

moisture deficit, dominates the occurrence of hot extremes.

## 2 Methodology

### 2.1 Data and metrics

The number of hot days is defined as the number of days per specific time interval (e.g., month, season, year) with a surface air temperature at 2 m height above the 90th-percentile. The percentile value is based on the distribution of the corresponding

five consecutive days (e.g., to determine if a 15[th] June is a hot day, the 90th percentile is based on all days of 13[th]-17[th] June) of the entire temperature time series (Mueller and Seneviratne, 2012). In this study, the number of hot days in the hottest month (NHD) is examined based on the ECMWF reanalysis ERA-Interim daily maximum temperature data (Dee et al., 2011). For each grid cell, the hottest month is determined based on the monthly average daily maximum temperature for the time series 1985-2015. Its geographical distribution is shown in Fig. 1.

Atmospheric $CO_2$ concentration is represented by an annual $CO_2$ concentration time series averaged from monthly data (Keeling et al., 2001) of the Mauna Loa, Hawaii station. The data are provided by the Scripps Institution of Oceanography, USA.

A reconstructed GRACE TWS dataset (Humphrey et al., 2017) from 1985 to 2015 provided by the Institute for Atmospheric and Climate Science, Eidgenössische Technische Hochschule Zurich (IAC ETH) is applied in this study. GRACE TWS has

been decomposed into "approximate" ($A1$, $A2$, $A3$, $A4$) and "detail" ($D1$, $D2$, $D3$, $D4$) components by a wavelet method following Andrew et al. (2017). The structure of wavelet decomposition is shown in Fig. 2, taking a grid cell in Australia (30.5





S, 130.5 E) as an example. The sum of a series of approximate and the corresponding detail components equals to the raw signal (e.g., raw signal=$D1+D2+A2=D1+D2+D3+A3$). Each decomposition level represents a certain time scale: $D1$ (2-month), $D2$ (4-month), $D3$ (8-month), and $A4$ and $D4$ (≥ 16-month). Those decomposed components of different temporal scales reflect

temporal dynamics of moisture at different depths. This is on the basis of the understanding that moisture at various soil depths has different response times to the climate system (Andrew et al., 2017; Chen et al., 2019). The original GRACE TWS data from 2003 to 2016 (Watkins et al., 2015; Wiese et al., 2019) provided by the Jet Propulsion Laboratory (JPL) are also applied in this study. Although the data period is too short to provide reliable statistical analysis, it provides a comparison for the reconstructed GRACE TWS.

For relating the occurrence of hot extremes to soil moisture deficit, SPI (calculated from GPCC Reanalysis precipitation data, Schneider et al., 2015) and a land surface model derived soil moisture product (GLDAS_NOAH10_M.2.1, Rodell et al., 2004; Rui, 2011) are compared with GRACE TWS. GLDAS_NOAH $\theta$ and GRACE TWS are correlated with NHD in the concurrent month. 3-month SPI characterizing the precipitation deficits accumulated in the previous two months together with the hottest month itself (McKee et al., 1993) is applied, since soil moisture in the hottest month includes contributions from infiltration

of precipitation in previous months. A 1°×1° spatial resolution is adopted for all datasets used in this study. Although the chosen resolution might be coarse for resolving detailed patterns of the $\theta$-NHD relationship, it allows to investigate land surface and atmosphere coupling at synoptic scale, where water and heat exchanges between large air mass and land surface.

## 2.2 Statistical analysis methods

Relationships between NHD and the potential predictor variables are examined by the Pearson linear correlation. The $t$ test

statistic is used to evaluate the statistical significance of the correlation coefficient ($r$). For testing the linear relationship between time-series grid datasets, erroneous rejection of null hypothesis inevitably happens at individual grid cells for several reasons as described in Wilks (2016), leading to false discovery of significant relationship. To address this problem, the threshold for significant $p$-values (normally 0.05) should be adjusted to control the False Discovery Rate (FDR). This adjustment is done based on the distribution of $p$-values of all grid cells and a prescribed parameter $\alpha_{FDR}$ which controls the

level of the False Discovery Rate. An $\alpha_{FDR}$ of 0.1 (=2$\alpha_{global}$, where $\alpha_{global}$ =0.05) should be used for gridded atmospheric data, as it often has strong spatial correlation (Wilks, 2016). After this adjustment, the in this study used threshold values for testing significant linear relationships are 0.0092 for NHD-$CO_2$, 0.0241 for NHD-SPI, 0.0235 for NHD-GLDAS_NOAH $\theta$, and 0.0191 for NHD-TWS.

Stepwise multiple linear regression (Draper and Smith, 1998; Clow, 2010) is used to determine the significant (5% significance

level assessed by an F-test) predictor variable in explaining NHD temporal variability for each grid cell. Next, the dominance analysis approach (Azen and Budescu, 2003) is applied to compare the relative importance (percentage of contribution is indicated by the coefficient of determination ($R^2$)) of those selected variables to identify the dominant influencing factor. In the dominance analysis, the overall $R^2$ of a predictor variable is calculated from all the possible subset regression models. The



predictor with highest conditional dominance (averaged from all sub-models) is identified as the largest contributor (dominant influencing factor) (Budescu, 1993; Azen and Budescu, 2003; Nimon and Oswald, 2013).

# 3 Results and discussion

## 3.1 Relationship between atmospheric CO2 concentration and hot extreme occurrence

Previous studies show that intensity, frequency, and duration of hot extremes are increasing (e. g., Perkins et al., 2012; Kim et al., 2016; Johnson et al., 2018). In this study, NHD also shows an upward trend consistent with the increasing $CO_2$ concentration in the past 31 years (Fig. 3 (a)). The correlation coefficient ($r$) between the annual NHD and the $CO_2$ concentration at each grid cell is mapped in Fig. 3 (b). The relationships are particularly strong in the tropics and parts of North America, East Europe, and Central and East Asia. Regions with significant positive correlation account for 11.3% of the land area in total.

## 3.2 Global θ-NHD relationship based on decomposed TWS

Correlations between NHD versus SPI, GLDAS_NOAH $\theta$, and TWS during 1985-2015 are compared in Fig. 4 (a-c). Similar spatial patterns of global $\theta$-NHD relationship are observed. Strong $\theta$-NHD relationships occur in most of the Americas, Europe, Australia, South Africa, East Asia, and Southeast Asia, which cover almost all of the areas with strong land-atmosphere coupling as identified in previous studies (e.g., Koster et al., 2006; Miralles et al., 2012; Schwingshackl et al., 2017; Donat et al. 2017, Chen et al., 2019). Significant negative correlations between NHD and SPI are observed for 25.5% of the land area, while NHD and GLDAS_NOAH $\theta$ are significantly correlated for 17.3% of the land area. This difference may be due to the fact that SPI, being used as a soil moisture proxy, reflects its influence on air temperature by soil-moisture dependent latent heat processes (evaporation and transpiration), in addition, it may reflect precipitation and air temperature coupling through weather systems (e.g., evaporation of rain water draws heat out of the near surface air).

The total area of significant $\theta$-NHD relationship increases, from 21.7% when the total terrestrial water storage is used, to 29.9% when the optimal decomposed TWS component at each grid cell is correlated to NHD (Fig. 4 (c-d)). This is likely because that a part of the terrestrial water storage is not directly accessible for evapotranspiration. From all soil moisture proxies, the decomposed GRACE TWS covers the largest land area with significant negative correlation with NHD (Fig. 4 (a-d)). It should be noted that only one TWS sub-component is used for Fig. 4 (d). The sum of all decomposed TWS components is expected to have a higher explanatory power for NHD temporal variability over a larger area.

Results shown in the first column of Fig. 4 are based on the reconstructed GRACE TWS dataset (1985-2015). We did the same analysis for the original GRACE TWS data from 2003 to 2016 for comparison, and the results are shown in the second column of Fig. 4. Based on the 14-year data, strong $\theta$-NHD relationships are also spatially distributed in most of the Americas, Europe, Australia, South Africa, East Asia, and Southeast Asia. In addition, the decomposed TWS (Fig. 4 (h)) shows significant correlation with NHD over a larger area than SPI (Fig. 4 (e)) and GLDAS_NOAH $\theta$ (Fig. 4 (f)). Although the period of





available original GRACE TWS data is relatively short at present, its contribution to this and related research will increase
with the accumulation of data and improved GRACE resolution in the future.

A significant negative $\theta$-NHD correlation can reflect a causal relationship between hot extremes and soil moisture deficit,
either way. If the correlation would indicate that high temperatures reduce soil moisture by increasing evapotranspiration, it
would be more likely to happen via evaporation of surface moisture, rather than transpiration from deep root zone. This is
because in the mid and low latitudes where the strong $\theta$-NHD correlation occurs (Fig. 4), temperature in the hottest month is
unlikely a limited factor for plant growth (Nemani et al., 2003). Under such a condition an increase in temperature does not
increase transpiration. Indeed, it more likely reduces transpiration due to stomatal responses to an increase in vapour pressure
deficit with temperature (Whitley et al., 2009; Wang et al., 2014). Since previous studies (Hirschi et al., 2014; Chen et al.,
2019) suggested that the coupling between air temperature and surface moisture is weak globally, the strong $\theta$-NHD
relationship more likely reflects the impact of soil moisture deficit on hot extremes. Indeed, it has already been suggested that
hot extremes in Europe (Hirschi et al., 2011), Australia (Herold et al., 2016), and parts of the Americas and South Africa
(Mueller and Seneviratne, 2012) are amplified by moisture deficit. In addition, a similar spatial pattern of significant NHD-
SPI correlation to those between NHD and other soil moisture indicators (Fig. 4), supports that the observed negative NHD
correlation reflects soil moisture deficit enhanced occurrence of hot extremes.
Fig. 5 shows the explanatory power (reflected by the regression $R^2$) of different soil moisture proxies for NHD variability
(1985-2015). The result is consistent with what is shown in Fig. 4 that SPI has stronger explanatory power for NHD variability
for a larger land area than GLDAS_NOAH $\theta$. The decomposed TWS shows the highest $R^2$ among all soil moisture proxies.
For testing the improvement, the adjusted $R^2$ is applied, which takes into account the total number of explanatory variables (up
to 5 for the decomposed TWS vs. 1 for other proxies) by including a penalty for having additional variables in the regression
analysis. The decomposed TWS shows significant adjusted $R^2$ for 33.4% of the land area. For 72.0% of this area the average
adjusted $R^2$ increases to 0.24 compared to the raw TWS average adjusted $R^2$ (0.09). For 28.0% of this area the raw TWS shows
a slightly higher average adjusted $R^2$ (0.30) than that of the decomposed TWS (0.24). This result sheds light on the potential
of decomposed GRACE TWS for hot extreme prediction.

Areas having significant negative correlation between NHD and decomposed TWS components during 1985-2015 are shown
in Figure 6. Those significant regions located in the central part of North America, the eastern part of South America, South
Africa and South Asia have deeper plant rooting depths (Fan et al., 2017), where interannual variability (D4 and A4) of TWS
seems to be more important than its seasonal variability (D1-D3) in explaining NHD temporal variability. This implies that
plant water uptake from deeper soil plays an important role in $\theta$-NHD coupling. However, D4+A4 also show stronger
correlation than D1+D2+D3 with NHD in areas without deep roots, including the northern and southeastern parts of South
America, the southwestern part of North Asia, and Southeast Asia. This is because those regions have shallow groundwater
table depth (Fan et al., 2013). It implies that in areas where groundwater is shallow, groundwater dependent ecosystems may
contribute to heat mitigation, which is worthy of future investigation.





### 3.3 Relative importance of increased atmospheric CO2 concentration and local moisture deficit for hot extreme occurrence

As shown above, both atmospheric $CO_2$ concentration and soil moisture have influence on the occurrence of hot extremes. This study aims to reveal, which factor, increased atmospheric $CO_2$ concentration or local moisture deficit, dominates hot extreme occurrence, and how this dominance varies spatially during the study period 1985-2015. The decomposed TWS is adopted to represent soil moisture in the dominance analysis. The results are mapped in Fig. 7, only the grid cells where the total explanatory power of atmospheric $CO_2$ concentration and decomposed TWS is over 95% significance level are

highlighted in color. That means in areas with gray colored grid cells other factors than increased atmospheric $CO_2$ concentration or local moisture deficit, may have a stronger influence on hot extreme occurrence. These factors may include variables such as ocean-atmosphere dynamics (e.g., Lorenzo and Mantua, 2016) and land use changes (e.g., Luo and Lau, 2017). During 1985-2015, for 18.2% of the land area with significant regression $R^2$, the occurrence of hot extremes is dominated by increased atmospheric $CO_2$ concentration. In most previously identified regions with strong land moisture and

air temperature coupling, including the northern areas of South America, the southern regions of North America, South Africa, West Europe, Southeast Asia, parts of East Asia and Australia, local moisture deficit shows very strong explanatory power for NHD temporal variability. It appears that local moisture deficit dominates the hot extreme occurrence for 40.6% of the land area with significant regression $R^2$. It is interesting that by comparing the distribution of high-income and low-middle-income countries (Fig. 8) we can roughly see that hot extremes in most low-middle-income countries located in Africa, Asia, and

South America are more sensitive to the factor of increased atmospheric $CO_2$ concentration than to local moisture deficit, while in high-income areas, such as West Europe, North America and Australia, this is opposite.

### 4 Conclusions

This study identifies which factor, increased atmospheric $CO_2$ concentration or local moisture deficit, is dominant in influencing the temporal occurrence of hot extremes at a global scale during 1985-2015. In parts of Africa, South America and

Asia, the occurrence of hot extremes is more sensitive to increased atmospheric $CO_2$ concentration than other areas. Local moisture deficit dominates hot extreme occurrence in regions with a total area twice as large as dominated by increased atmospheric $CO_2$ concentration during the 31-year period investigated here, which is an important new realization. These regions, i.e., North America, West Europe, Australia, Southeast Asia, and South Africa are previously identified as having strong land-atmosphere coupling, which influences the moisture deficit-hot extreme links. Hence, those regions may mitigate

some hot extremes by addressing the increasing moisture deficit by e.g. adaptive land management. We note that under continuing increase of greenhouse gas forcing, hot extremes are expected to be dominated by increased $CO_2$ concentration over larger areas in the future. Hence, global measures for reducing emissions are essential in combating current and future expansion of hot extremes.



The dominance analysis approach is applied to quantify relative importance of increased atmospheric $CO_2$ concentration and
local moisture deficit to the occurrence of hot extremes during 1985-2015. The application of decomposed GRACE TWS in
estimating the global distribution of hot extremes is here for the first time presented, it shows larger areas with significant $\theta$-
NHD relationships and higher regression $R^2$ in examining the occurrence of hot extremes than the other commonly used soil
moisture proxies SPI and land surface model derived product. It suggests the potential of decomposed GRACE TWS as a
useful soil moisture proxy in examining moisture-heatwave coupling.

**Code availability.**

All computer codes used to analyse the data and produce the plots are available from the authors upon request.

**Data availability.**

Daily    maximum    air    temperature    datasets    for    calculating    NHD    (ERA-Interim)    are    available    from
http://apps.ecmwf.int/datasets/data/interim-full-moda/levtype=sfc/; $CO_2$ concentration of Mauna Loa, Hawaii station provided
by GES DISC are downloaded from https://scrippsco2.ucsd.edu/data/atmospheric_co2/primary_mlo_co2_record.html; the
original    GRACE    TWS    data    (2003-2016)    provided    by    JPL    are    available    from
https://podaac.jpl.nasa.gov/dataset/TELLUS_GRAC-GRFO_MASCON_GRID_RL06_V2; the reconstructed GRACE TWS
data (1985-2015) provided by the Institute for Atmospheric and Climate Science, Eidgenössische Technische Hochschule
Zurich (IAC ETH) are available from http://rossa-prod-ap21.ethz.ch/delivery/DeliveryManagerServlet?dps_pid=IE5766472;
monthly    precipitation    datasets    (GPCC)    for    calculating    3-month    SPI    are    downloaded    from
https://www.esrl.noaa.gov/psd/thredds/catalog/Datasets/gpcc/full_v7/catalog.html; the model derived soil moisture products
(GLDAS_NOAH    $\theta$)    are    available    from
https://disc.gsfc.nasa.gov/datasets/GLDAS_NOAH10_M_V2.1/summary?keywords=GLDAS.

**Author contribution.**

Ajiao Chen: Conceptualization, Formal analysis, Writing-Original draft preparation; Huade Guan: Conceptualization,
Supervision, Writing-Reviewing and Editing; Okke Batelaan: Supervision, Writing-Reviewing and Editing

**Competing interests.**

The authors declare that they have no conflict of interest.

**Acknowledgements**

We appreciate Dr Xinguang He from Hunan Normal University, China for providing Wavelet Decomposition Matlab code,
and Dr Stephen Broomell, Florian Lorenz, and Nathaniel E. Helwig from Carnegie Mellon University for providing
Dominance Analysis Matlab code. The first author also appreciates the financial support from the China Scholarship
Council, Ministry of Education, China, and Flinders University, Australia.





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


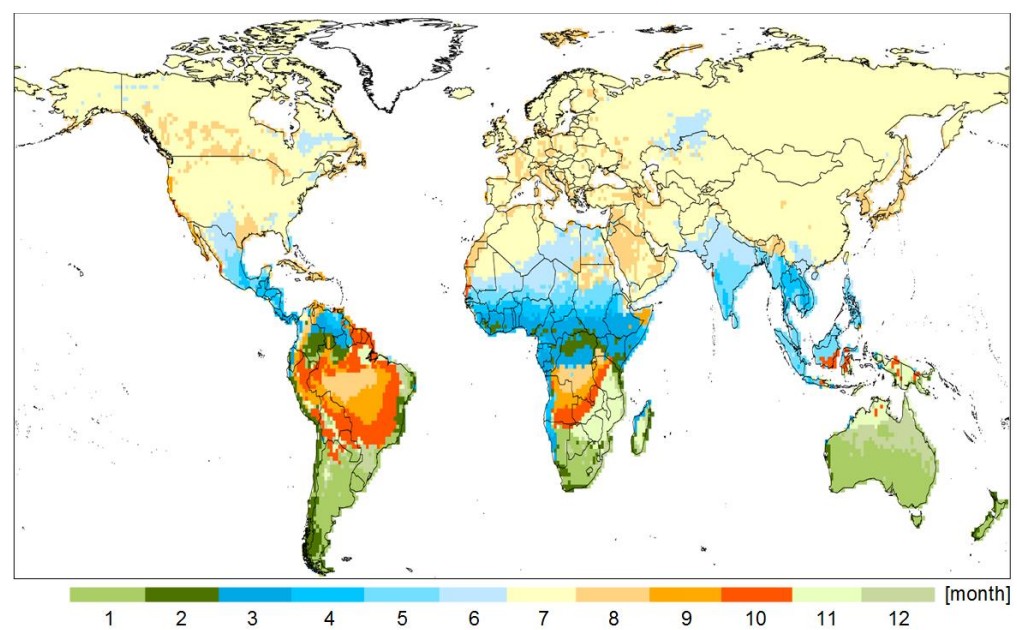

**Figure 1: Geographical distribution of most frequent occurring hottest month for the period 1985-2015.**

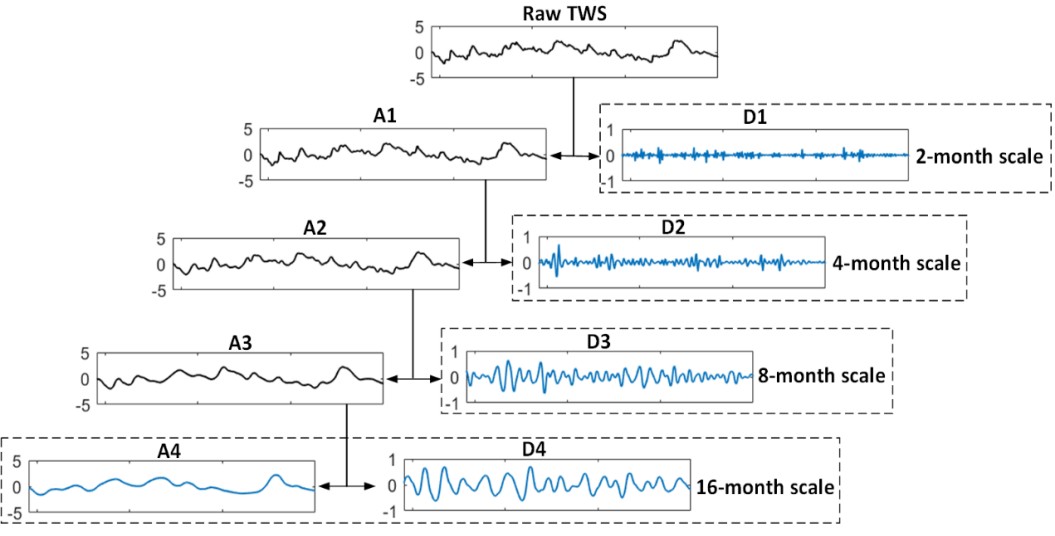


**Figure 2: The structure of a discrete wavelet decomposition (an example from a grid cell in Australia (30.5 S, 130.5 E)).**

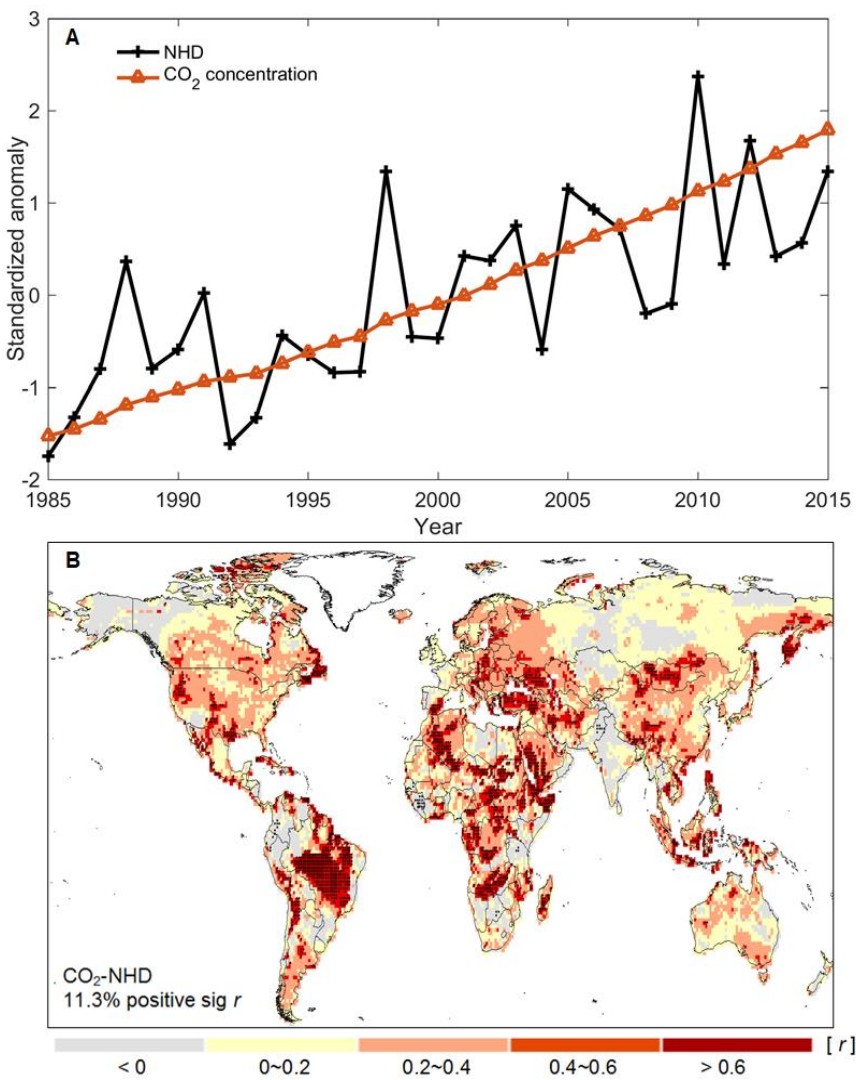

**Figure 3: (a)** Standardized anomaly of global average NHD (land regions only) and atmospheric CO₂ concentration. The standardized anomalies are calculated based on the mean and standard deviation derived from the full period 1985-2015. **(b)** Correlation coefficients (*r*) between annual NHD and CO₂ concentration at each grid cell. Significant levels are denoted by black dots. No data is available for land area marked in white.





**Figure 4: Correlations between NHD and (a) SPI; (b) GLDAS_NOAH θ; (c) raw TWS; and (d) the maximum *r* value of NHD versus**
**any of the decomposed TWS components during 1985-2015 (based on reconstructed GRACE TWS data). Significant levels are denoted by black dots. No data is available for land area marked in white. The second column is same as the first column but for the period 2003-2016 (based on JPL GRACE TWS data).**



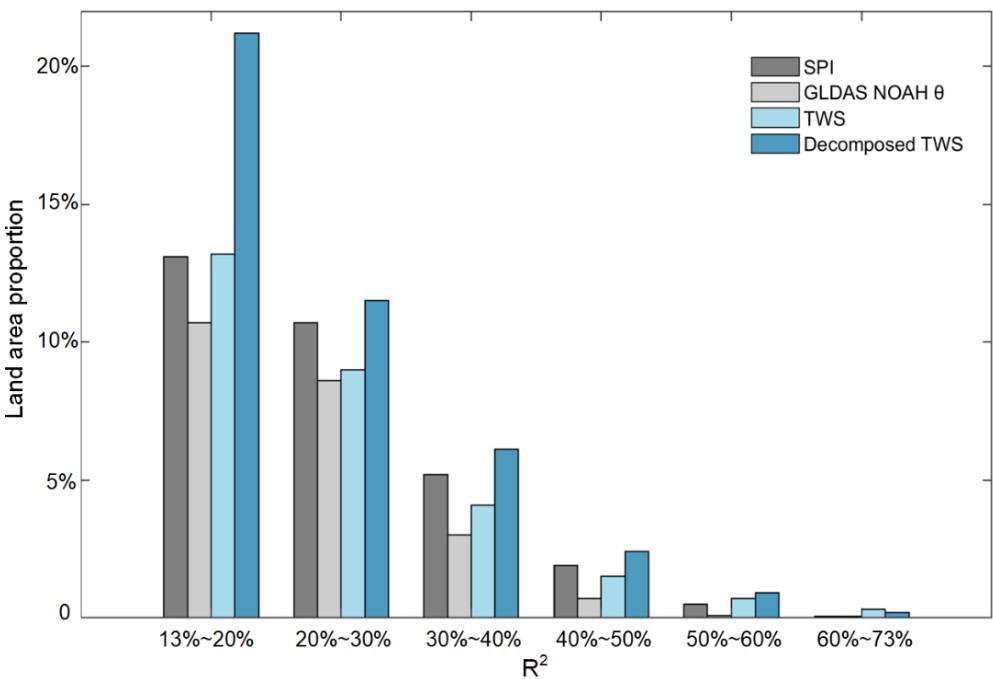

**Figure 5: Histogram of the explanatory power (significant regression $R^2$) on NHD variability by using SPI, GLDAS_NOAH $\theta$, raw TWS, and decomposed TWS during 1985-2015.**

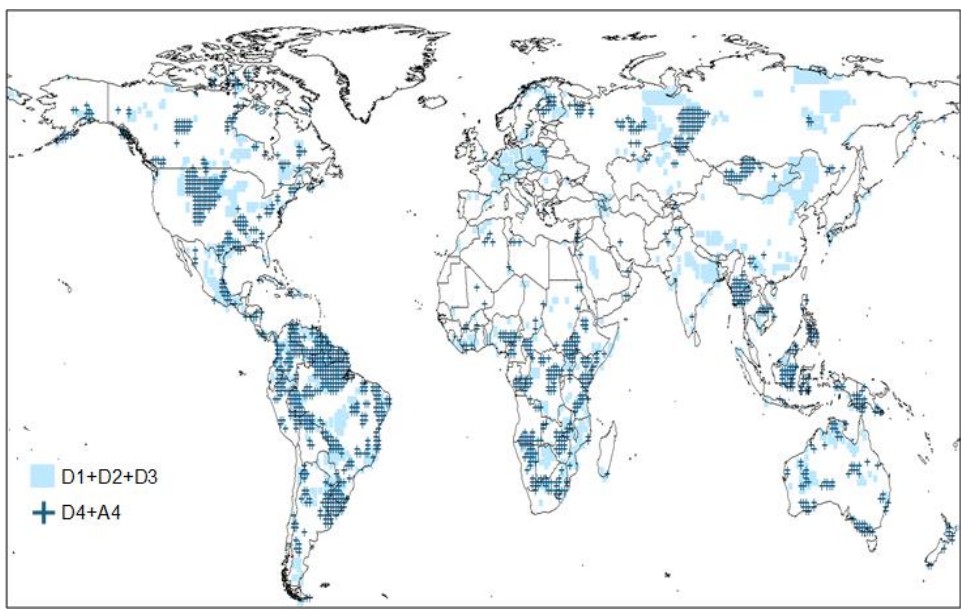

**Figure 6: Areas where significant negative correlation exists between NHD and moisture at shallower soil depth (D1+D2+D3) and deeper soil depth (D4+A4) represented by wavelet decomposition levels of TWS (1985-2015).**



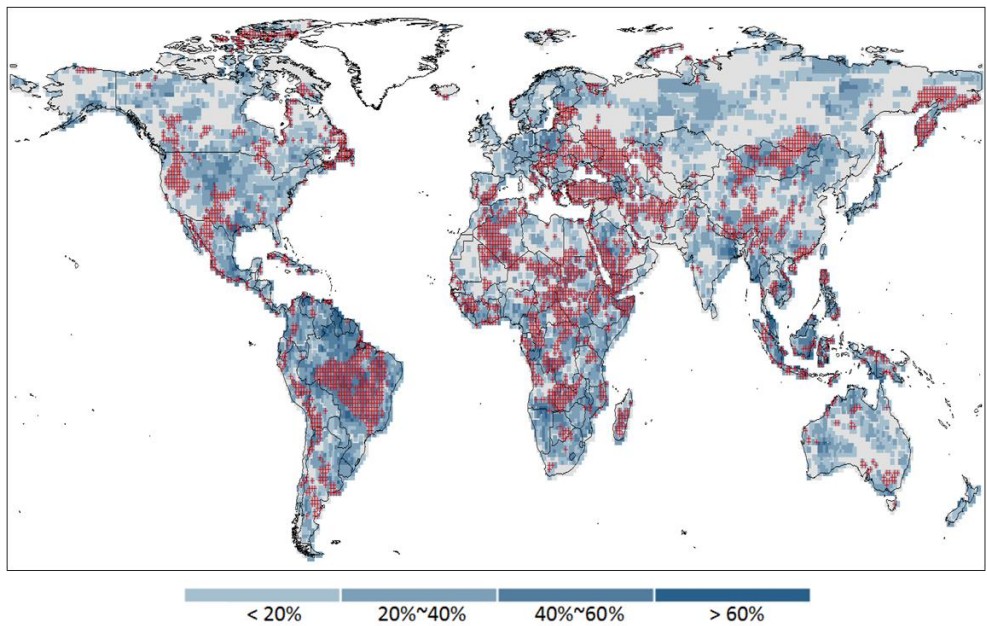


**Figure 7: Spatial pattern of the total explanatory power of the joint influence of atmospheric $CO_2$ concentration and soil moisture on hot extreme occurrences. The occurrence of hot extremes in areas marked with red cross symbols is dominated by increased atmospheric $CO_2$ concentration, while in areas with blue colors it is dominated by local moisture deficit. Hot extreme occurrence in the grey areas are not significantly associated with either atmospheric $CO_2$ concentration or soil moisture during the study period 1985-2015.**


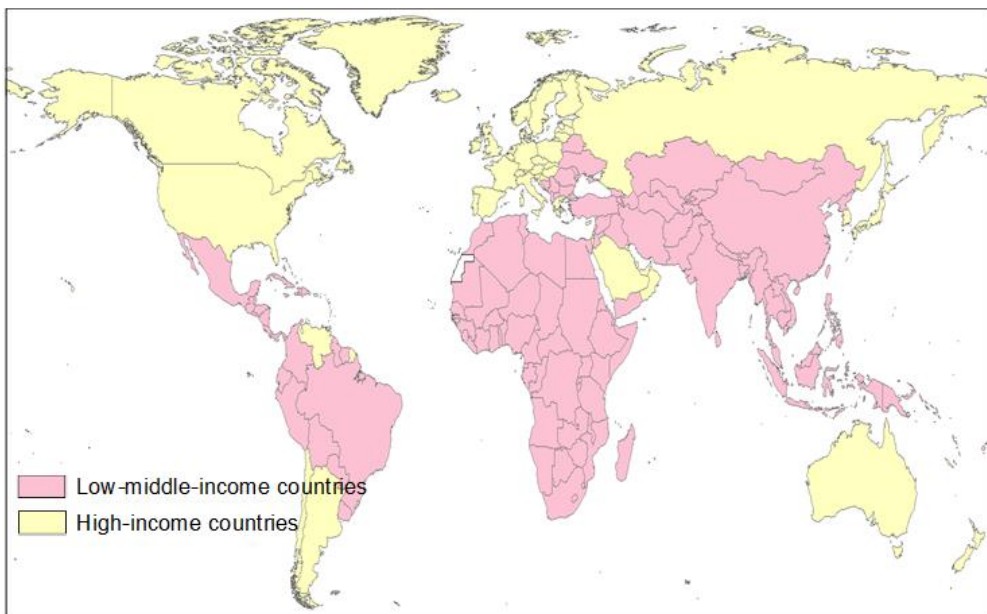

**Figure 8: Distribution of high and low-middle-income countries. Income level information is available from the World Bank, (https://datahelpdesk.worldbank.org/knowledgebase/articles/906519).**