# Peer review of "Relative importance of increased atmospheric CO2 concentration and local moisture deficit to hot extremes"

_Hydrology and Earth System Sciences, 2020_

## Referee Comment (RC1) · Ryan Teuling (Referee) · 19 Oct 2020

I have taken a bit more time than I normally do to review this work, because the study is interesting but also because I felt there might be some potential issues with the analysis that I wanted to get clear. The manuscript by Chen and co-workers focusses on the occurrence of hot extremes and their relation to global warming on the one hand, and local soil moisture conditions on the other. This is relevant because both factors have been shown in previous studies to play a role in the occurrence of, and trend in, hot days. Simply put, the authors analyze where the correlation between heatwave days and global $CO_2$ levels is higher, and where the correlation with local soil moisture deficit

is higher. The main question for me as a reviewer is whether such a comparison makes sense, and is fair.

I feel there are several issues with this approach, related to the variables selected for the analysis, as well as with the data used to calculate the correlations. Firstly, the selection of yearly-average, global $CO_2$ levels is somewhat arbitrary. Yes, they drive global warming, but nobody would claim that the yearly variation of $CO_2$ would directly influence hot days at any particular location. The link is simply too indirect and weak, since it depends not only on how $CO_2$ levels influence global temperatures, but also circulation. $CO_2$ is also only one of the greenhouse gasses. In a way, it would be more logical to use global average temperature deviations instead, since these at least reflect the effect of ENSO and other climate variability that is known to affect the occurrence of hot extremes. But even there the weak is link, since in many regions the year-to-year variability in heatwaves is much more closely linked to indices such as ENSO, NAO etc. than it is to global $CO_2$ or local soil moisture. Again I am not talking here about the fact that heatwaves increase with global warming, and that this increase correlates with global $CO_2$, but the point is that the absolute value of the correlation coefficient is meaningless and should not be compared to other correlations because it results from arbitrary choices.

A second problem is caused by the data used. From what I was able to find about the GRACE TWS dataset used, it seems that the different timeseries have been detrended. This is problematic in the view of the main research question, because any temporal trend in storage due to climate change will now be attributed to changes in $CO_2$ only. This also shows that $CO_2$ and soil moisture impacts will be difficult if not impossible to separate from observations only, since the two are not independent. A second problem is that the different decomposed signals should be interpreted with care. Even if they do reflect soil moisture at different depths as the authors claim, it will be unlikely that any storage with response timescales of several months will significantly impact energy balance partitioning at the land surface, and air temperature. In case of the significant

correlations found by the authors, Occam's razor should be applied first: the more simple explanation is that both are affected by persistence in the atmospheric forcing signal due to persistence in atmospheric circulation. Only is this can be excluded as explanation should the correlation be interpreted as the result of direct soil moisture effect on air temperature.

In conclusion, I believe the separation between $CO_2$ and soil moisture effects on hot extremes cannot be done from correlation analysis on observations along the lines of the analysis presented here, but would require dedicated experiments with coupled climate models. Since in my view the conclusions of this work on the relative importance on both processes are not sufficiently justified by the evidence presented, I have to conclude that the current version of this work is unfortunately not suitable for final publication in HESS.

---

## Author Comment (AC1) · 27 Oct 2020

**Authors' response to interactive comment of the Referee #1**

**Black text: Referee comment**
**Blue text: Authors' response**

We greatly appreciate your comments and suggestions for improving our manuscript. Our response and revision plans are provided below.

1. I have taken a bit more time than I normally do to review this work, because the study is interesting but also because I felt there might be some potential issues with the analysis that I wanted to get clear. The manuscript by Chen and co-workers focusses on the occurrence of hot extremes and their relation to global warming on the one hand, and local soil moisture conditions on the other. This is relevant because both factors have been shown in previous studies to play a role in the occurrence of, and trend in, hot days. Simply put, the authors analyze where the correlation between heatwave days and global $CO_2$ levels is higher, and where the correlation with local soil moisture deficit is higher. The main question for me as a reviewer is whether such a comparison makes sense, and is fair.

Thanks for spending time on reviewing our work, and we are very glad that you are interested in this study.

To address the issue that you are concerned with, we would like to explain that our conclusion with respect to which factor is more important in influencing hot extremes is not based on comparing the absolute value of correlation coefficients. We applied the dominance analysis (Azen and Budescu (2003) (https://doi.org/10.1037/1082-989X.8.2.129)) to compare the relative importance of different influencing factors to the occurrence of hot extremes. The dominance analysis method became popular in social science, it can compare the relative importance of predictors in multiple regression. Recently, this method has also been applied in natural sciences. Even though the predictors are correlated, the dominance analysis can estimate their relative importance. For example, Asoka et al., (2017) (DOI: 10.1038/NGEO2869) used it to investigate the relative contribution of monsoon precipitation and pumping to changes in groundwater storage in India. We hope this explanation addresses your concerns, and in a revised manuscript we will provide more detail of the dominance analysis as follows:

"Stepwise multiple linear regression (Draper and Smith, 1998; Clow, 2010) is used to determine the significant (5% significance level assessed by an F-test) predictor variable in explaining NHD temporal variability for each grid cell. Next, the dominance analysis approach (Budescu, 1993; Azen and Budescu, 2003) is applied to compare the relative importance of those selected variables. The total variance among a set of predictors can be fully partitioned by dominance analysis even if the predictors are correlated (Vize et al., 2019). The issue of collinearity among predictors is addressed by examining the unique variance accounted for by the predictor across all possible regression sub-models involving the predictor. Dominance analysis is completed through an exhaustive set of pairwise comparisons among the predictors. The comparisons can be examined by three types of dominance: complete dominance, conditional dominance, and general dominance (Nimon & Oswald, 2013). To be completely dominant, a predictor must account for a greater amount of outcome variance than another predictor across every sub-model comparison. The conditional dominance of different predictors is conditional on what

sub-model level is being examined. We applied the general dominance in this study, which is determined by taking the average amount of variance accounted for by a predictor across all sub-models and comparing it to other predictors. General dominance weights can be calculated for each predictor in a set and represent the relative proportion of $R^2$ attributable to a predictor."

2. I feel there are several issues with this approach, related to the variables selected for the analysis, as well as with the data used to calculate the correlations. Firstly, the selection of yearly-average, global $CO_2$ levels is somewhat arbitrary. Yes, they drive global warming, but nobody would claim that the yearly variation of $CO_2$ would directly influence hot days at any particular location. The link is simply too indirect and weak, since it depends not only on how $CO_2$ levels influence global temperatures, but also circulation. $CO_2$ is also only one of the greenhouse gasses. In a way, it would be more logical to use global average temperature deviations instead, since these at least reflect the effect of ENSO and other climate variability that is known to affect the occurrence of hot extremes. But even there the weak is link, since in many regions the year-to-year variability in heatwaves is much more closely linked to indices such as ENSO, NAO etc. than it is to global $CO_2$ or local soil moisture. Again I am not talking here about the fact that heatwaves increase with global warming, and that this increase correlates with global $CO_2$, but the point is that the absolute value of the correlation coefficient is meaningless and should not be compared to other correlations because it results from arbitrary choices.

We agree that many factors (including atmospheric circulation patterns) together determine hot extreme occurrence. Global warming and soil moisture examined in this study are two of them. You are concerned that the links between the occurrence of hot extremes and the two factors are weak. We would like to address this concern first, followed by explanation of why global $CO_2$ concentration is used as proxy data for global warming in our analysis, and why the circulation pattern is not directly included in our analysis.

This study examines number of hot days in summer, which tend to occur under clear sky conditions during daytime. Standard 2-metre-height air temperature data were analyzed here. Physically, the heat source for near-surface air during daytime is directly from sensible heating of the underlying surface. Local soil moisture conditions strongly influence partitioning of net radiation on the surface into sensible heat and latent heat. Thus, it is reasonable to expect an association between number of hot days and soil moisture. On the other hand, global warming increases heat storage in the atmosphere, which tends to increase the likelihood of hot day occurrence. Thus, we can expect an association between number of hot days and a global warming variable.

We agree with you that global average temperature deviation is a rational proxy to be used in this study, which can reflect effects of both global warming and climate variability on the occurrence of hot extremes. We tried deviation of global average temperature before, the result looks very similar to what $CO_2$ concentration shows (see the figure shown below, dominance analysis based on temperature (a) vs. $CO_2$ (b)).

[Figure]

The consistency between the two maps is not surprising, as the major component of greenhouse gas, $CO_2$ is reported to have the highest positive radiative forcing (73%) of all the human-influenced climate drivers (IPCC (2013)).

We agree that when and where a hot extreme event occurs is determined by the atmospheric circulation (weather), which may include climate oscillation (e.g., ENSO) effects. But this process is in general chaotic, beyond the control of human society. $CO_2$ concentration as well as soil moisture are more likely affected by human activities. Thus, results in this study are expected to provide practical advice for society, i.e., global measures on reducing greenhouse gas emission and adaptive land management in some regions with increasing moisture deficit, to mitigate heatwaves.

We appreciate your comment, which reminds us that clearer explanation on why we selected those variables is needed in our manuscript. Besides, in order not to mislead readers that we ignored the strong effect of atmospheric circulation on hot extremes, we will clearly state that the aim of this study is to compare the relative importance of $CO_2$ concentration and soil moisture to the occurrence of hot extremes instead of identifying the dominant driver among all influencing factors.

3. A second problem is caused by the data used. From what I was able to find about the GRACE TWS dataset used, it seems that the different timeseries have been detrended. This is

problematic in the view of the main research question, because any temporal trend in storage due to climate change will now be attributed to changes in $CO_2$ only. This also shows that $CO_2$ and soil moisture impacts will be difficult if not impossible to separate from observations only, since the two are not independent. A second problem is that the different decomposed signals should be interpreted with care. Even if they do reflect soil moisture at different depths as the authors claim, it will be unlikely that any storage with response timescales of several months will significantly impact energy balance partitioning at the land surface, and air temperature. In case of the significant correlations found by the authors, Occam's razor should be applied first: the more simple explanation is that both are affected by persistence in the atmospheric forcing signal due to persistence in atmospheric circulation. Only is this can be excluded as explanation should the correlation be interpreted as the result of direct soil moisture effect on air temperature.

Yes, all 'detail' components (D1-D4) of GRACE TWS have been detrended, but in all 'approximate' components (A1-A4) the trends are contained. Since D1-D4 and A4 (the sum of all these components equals to the raw TWS signal) are all included in dominance analysis in this study, variation and trends in heatwave days can be attributed to both $CO_2$ concentration and soil moisture. In fact, the two variables Global $CO_2$ and GRACE TWS adopted in this study are mostly independent. Here we provide a map (see the figure shown below) to show the correlation between $CO_2$ and TWS (only $p$ value is shown). There are very few grid cells (marked in red) showing significant correlation ($p=0.05$) between $CO_2$ and soil moisture. This result and the relevant explanation will be added into the revised manuscript. We believe this can address your concerns about the independence between the two variables.

[Figure]

Regarding your concern on how we use wavelet decomposed TWS time series in our analysis, and particularly your questioning on the relationship of multiple-month signal and heat extreme occurrence, we would like to provide the following explanation.

The approach of using decomposed GRACE TWS to estimate various water storage components was proposed by Andrew et al., (2017) (http://dx.doi.org/10.1016/j.jhydrol.2017.06.016). The method was demonstrated based on time series of soil moisture measurements from near surface to a depth of six meters (see the figure

shown below). The depth integrated total soil water storage is mathematically used to mimic the behavior of GRACE raw TWS. This "raw TWS" is then decomposed into sub-time series at discrete time scales. These decomposed time series are then used to compose shallow and deep soil moisture time series. It is clear that even shallow soil moisture has a component of long-time scale, although its proportion is smaller than that in deep soil.

[Figure]

Copied from Andrew et al., (2017). Results using the wavelet decomposition and stepwise regression method for estimates of soil moisture at different depths. Plots a and b show the depth-integrated soil moisture vs. the shallow and deep layers. Plots c and d show the estimations and observations of soil moisture for the shallow and deep soil layers.

In addition, Andrew et al., (2016) (doi:10.5194/hess-2016-545) reported that vegetation responses to terrestrial moisture changes are of multiple months scales. Grassland-dominated areas are more sensitive to higher frequencies of moisture storage changes while plants with deeper rooting systems (e.g., forests) are more sensitive to moisture storage changes of longer time scales. Therefore, it is possible that moisture storage with response timescales of several months could impact energy balance partitioning at the land surface.

Based on the explanations above, we believe it is reasonable to examine the effect of soil moisture on air temperature. As you mentioned that the effect of atmospheric circulation on the occurrence of hot extremes could be stronger than effects of $CO_2$ concentration and soil moisture, it is necessary to clarify in our manuscript that the aim of this study is to compare the relative importance of the two variables we selected rather than identifying the dominant driver among all influencing factors. We will modify the relevant sentences in the revised manuscript.

4. In conclusion, I believe the separation between $CO_2$ and soil moisture effects on hot extremes cannot be done from correlation analysis on observations along the lines of the analysis presented here, but would require dedicated experiments with coupled climate models. Since in my view the conclusions of this work on the relative importance on both processes are not sufficiently justified by the evidence presented, I have to conclude that the current version of this work is unfortunately not suitable for final publication in HESS.

We agree that coupled climate models are very useful in dedicated numerical experiments and analyses. However, we believe that before these models are applied it is essential that data driven analyses are performed to learn from the data what the relative importance is of different variables. As we explained above, the dominance analysis method we used in this study, can quantify relative importance of predictors in a multiple regression. In addition, correlation between $CO_2$ and TWS signals are very weak as shown in our reply to your previous comment. We will add that figure and relevant descriptions into the revised manuscript in case readers have the same concerns as you.

---

## Referee Comment (RC2) · Ryan Teuling (Referee) · 28 Oct 2020

Here I want continue the discussion on the use of dominance analysis, and the variables used in the analysis. The authors provide the example of groundwater, where the technique was used to study the relative impacts of monsoon rainfall and pumping on the groundwater table dynamics. This example actually illustrates my point fairly well. In case of rainfall and pumping, nobody would argue against that these are the two main factors that control groundwater table dynamics in a rather direct, predictable and proportional way. This however is not the case when considering impacts of global CO2 and local soil moisture deficits. The link between global CO2 is

indirect, and opposite to the groundwater example nobody would argue that there is or should be a direct link between global $CO_2$ levels and HWD at any particular time and location. The same is true for soil moisture. Soil moisture is known to contribute to heatwave temperatures (e.g. Miralles et al., 2014, doi:10.1038/ngeo2141), but much of the local heat actually comes from advection driven by circulation patterns (e.g. Rasmijn et al., 2018, doi:10.1038/s41558-018-0114-0 and Schumacher et al, 2019, doi:10.1038/s41561-019-0431-6). In spite of the strong correlation, the contribution of local soil moisture to heatwave temperatures is important, but by no means dominant. The problem here is the classical pitfall that correlation is not causality. Soil moisture and temperature will be strongly negatively correlated in many regions simply because synoptic conditions leading to high temperatures (clear skies) are the same as those enhancing soil drying. But this does not mean that dry soils cause the high temperatures. In regions that are wet enough for ET not to become limited by soil moisture even during hot extremes, one would not expect soil moisture to impact temperature. By only using simple correlation, these regions will incorrectly be flagged as regions where soil moisture impacts temperature. To circumvent this, more complex coupling metrics have been developed that look for instance at anomalies in the surface energy balance (see Miralles et al. (2012) doi:10.1029/2012GL053703 among many others). The main factors in determining year-to-year variability of HWDs, like circulation indices, are not considered here. By only looking at correlation between variables that only weakly and indirectly impact HWDs, statistically significant results might be found, but that doesn't mean that they also provide new or meaningful insights.

---

## Author Comment (AC2) · 6 Nov 2020

**Authors' response to interactive comment of the Referee #2**

**Black text: Referee comment**
**Blue text: Authors' response**

Here I want to continue the discussion on the use of dominance analysis, and the variables used in the analysis. The authors provide the example of groundwater, where the technique was used to study the relative impacts of monsoon rainfall and pumping on the groundwater table dynamics. This example actually illustrates my point fairly well. In case of rainfall and pumping, nobody would argue against that these are the two main factors that control groundwater table dynamics in a rather direct, predictable and proportional way. This however is not the case when considering impacts of global $CO_2$ and local soil moisture deficits. The link between global $CO_2$ is indirect, and opposite to the groundwater example nobody would argue that there is or should be a direct link between global $CO_2$ levels and HWD at any particular time and location. The same is true for soil moisture. Soil moisture is known to contribute to heatwave temperatures (e.g. Miralles et al., 2014, doi:10.1038/ngeo2141), but much of the local heat actually comes from advection driven by circulation patterns (e.g. Rasmijn et al., 2018, doi:10.1038/s41558-018-0114-0 and Schumacher et al, 2019, doi:10.1038/s41561-019-0431-6). In spite of the strong correlation, the contribution of local soil moisture to heatwave temperatures is important, but by no means dominant. The problem here is the classical pitfall that correlation is not causality. Soil moisture and temperature will be strongly negatively correlated in many regions simply because synoptic conditions leading to high temperatures (clear skies) are the same as those enhancing soil drying. But this does not mean that dry soils cause the high temperatures. In regions that are wet enough for ET not to become limited by soil moisture even during hot extremes, one would not expect soil moisture to impact temperature. By only using simple correlation, these regions will incorrectly be flagged as regions where soil moisture impacts temperature. To circumvent this, more complex coupling metrics have been developed that look for instance at anomalies in the surface energy balance (see Miralles et al. (2012) doi:10.1029/2012GL053703 among many others). The main factors in determining year-to-year variability of HWDs, like circulation indices, are not considered here. By only looking at correlation between variables that only weakly and indirectly impact HWDs, statistically significant results might be found, but that doesn't mean that they also provide new or meaningful insights.

This is an interesting discussion. No doubt that this discussion will help to revise the next version of the manuscript.

We agree with you that synoptic weather systems, related to large-scale atmospheric circulations, play very important roles in influencing the occurrence of hot extremes. But the aim of this study is not looking for the dominant driver among all influencing factors. We aimed to identify the dominant one (more important one) between the two selected influencing factors (global $CO_2$ concentration and local soil moisture). The use of the word 'dominant' might be the root of your concern. We will replace the word 'dominant'/'dominates' by 'is more important' in next revision.

We have used both deviation of global mean annual temperature and global atmospheric $CO_2$

to approximate the global warming, resulting in very similar patterns (shown in our last response). $CO_2$ concentration was finally selected in the manuscript because it is something that human society can take action on. In addition, as we described in the previously submitted manuscript that "The observed global warming is considered extremely likely associated with anthropogenic influences, particularly greenhouse gas emission (IPCC, 2013). An increase in atmospheric $CO_2$ concentration as a consequence of emissions can cause an increase in extreme temperature (Min et al., 2013; Kim et al., 2016; Seneviratne et al., 2016; Baker et al., 2018).". We think it is reasonable to use $CO_2$ as one influencing factor in this study. However, if the editor thinks it is better to use global mean annual temperature, we are happy to do so.

The physical connection between soil moisture and hot extremes is becoming clear as Lisa Alexander explains in her article *Extreme heat rooted in dry soils*, (Alexander, 2011, doi: 10.1038/ngeo1045). We agree with you that the correlation between the two does not necessarily always points to a causal relationship in one direction or the other. Nevertheless, in the hottest month (for which the number of hot days is investigated here), the negative correlation between soil moisture and temperature is more likely to reflect the feedback of dry (primarily root-zone) soil to the atmosphere. Such feedback is expected to enhance the occurrence of hot extremes. Based on the mechanism that low soil moisture availability reduces evaporative cooling and increases atmospheric heating from sensible heat flux (Alexander, 2011), Mueller and Seneviratne (2012) (doi: 10.1073/pnas.1204330109) used correlation between hot days in the hottest month and 3-month SPI (a proxy for soil moisture) as coupling diagnostic to identify hot spots at a global scale. Those hot spots agree well with transitional climate regions (Koster et al., 2004 (doi: 10.1126/science.1100217); Seneviratne et al, 2010 (doi: 10.1016/j.earscirev.2010.02.004)) where soil moisture strongly constrains evapotranspiration variability and thus result in feedbacks to the atmosphere.

In our previous work, we did examine the other direction of coupling (i.e., hot temperature dries soil), similar as what you suggested. We investigated the correlation between temperature anomaly and remote sensed surface soil moisture and found very low correlations (see figure shown below). For some spots, the correlation is even positive, which is counterintuitive.

[Figure]

Figure R2.1. Correlation coefficients ($r$) between air temperature and surface soil moisture. Dots indicate that the corresponding $r$ has passed the significance test at 0.05 significance level.

It appears that although we have discussed the causal relationship between hot extremes and soil moisture deficit in the previously submitted manuscript, it is likely not clear enough. We will improve the relevant explanation in next revision.

We hope the above discussion will convince you that it is meaningful to investigate the *relative* importance of the two selected variables, global $CO_2$ concentration (or average temperature deviations) and soil moisture, to hot extremes. This does not at all mean that the circulation pattern is not important in the occurrence of specific hot extremes. Since the process of atmospheric circulation is beyond the control of human society, we selected two influencing factors (global warming and soil moisture deficit) that are more likely affected by human activities so that the corresponding results are expected to provide practical advice for society in mitigating heatwaves. We acknowledge that some descriptions in our previously submitted manuscript may not be clear, we will revise the relevant text to avoid misunderstanding.

---

## Referee Comment (RC3) · Anonymous Referee #2 · 17 Jan 2021

This research presents evidence on which driver, either the enhancement of CO2 molar fraction or the soil moisture deficit, dominates the temporal occurrence of hot extremes at the global scale. The methodology is based on applying a wavelet analysis (GRACE) to a long climatologically data set that ranges from 1985 until 2015. The findings identify which regions characterized by different ecosystems are influenced by soil moisture deficit and which ones by the atmospheric CO2 molar fractions. Although the findings are per se might be worth to be published, I found that the analysis and the writing of the paper is done in a hasty manner and therefore requires major revisions.

Below my recommendations:

1.- I think the representativeness of the statistical analysis should be placed in a better perspective and with clearer justifications. I am particular concern on how they attribute the correlations to the specific dominance of a driver. This requires a much in-depth elaboration throughout all the paper.

2.- I believe it is interesting to provide a global perspective, but I think the authors are in a position to provide more evidence at the regional scale. For instance, in Figure 3, they could select representative regions characterized by differentiated ecosystems and explain in more detailed the differences. At figure 3b, it is clear that the Amazonian basin has a contrast behavior in two areas that is also observed in Figure 7. Why? Could they please elaborate and provide more detail explanation supported by figures?

3.- Closely connected to this, I miss more in-depth explanations on the causality of their effects. There are some attempts to explain a connection of feedbacks (lines 131-133), but in the majority of the results the reader is left out . In my opinion, at section 3.3 (I will call it discussion) the authors have an unique opportunity to provide some diagrams that show the feedback relations and the effects of the enhancement of $CO_2$ or soil deficit in the hottest month.

4.- In my opinion, there is a driver that is missing in the discussion: the water vapour pressure deficit. What is the role played by this variable in enhancing the warming of the hot extremes? Could they calculate it also using their wavelet analysis and then relate it to their findings?

5- The wording throughout the article is casual and not very exact. Would it be possible to identify the important ecosystems like the tropical rain forest, temperate or boreal forests instead of mentioning the continents (Africa, South America,...)?

6- Figure 8 at section 3.3 appears out of the blue. In my opinion, it needs to be removed or much better embedded. As an alternative, un my opinion preferable, the extra space should be addressed to a more in-depth explanation of the cause-effects dominance of either enhanced $CO_2$ or soil moisture deficit on the hot extremes.

---

## Referee Comment (RC4) · Anonymous Referee #3 · 27 Jan 2021

The manuscript analyses the relative importance of global atmospheric CO2 concentration and local soil moisture for the number of hot days of the hottest month (NHD). Much of the manuscript is focused on the analysis of the relationship between decomposed GRACE TWS and NHD. I agree that this is an interesting and novel aspect at the core of the manuscript, which should potentially be in the title (e.g. Relevance of TWS deficit and increased atmospheric CO2 for hot extremes).

Overall, the manuscript is well-structured and clear. Nevertheless, it would be useful to further clarify and improve some aspects of the methodology and presentation of the results. It doesn't seem like the best approach to focus on a direct comparison of

moisture deficit and increasing CO2, given that changes in moisture deficit are also influenced by increasing CO2. Also, the computed correlations are taking into account both the trend and interannual variability. Looking separately at the trend and at the detrended interannual variability could be a good option. My hypothesis is that the trend in increasing CO2 correlates better with the trend in NHD, whereas the detrended interannual variability in NHD is clearly more related to the detrended interannual variability in local moisture deficit (the detrended variability in CO2 is likely very small). This can provide helpful context to better understand statements such as the conclusion in lines 180–182. One should not interpret these results as increasing CO2 is not too important for hot extremes; the fact that the correlation is not very high is likely because the detrended variability in CO2 is likely rather small and the study period relatively short.

Specific comments

1. In the first paragraph of the introduction, it could also be noted that "local moisture deficit" and "increased atmospheric CO2 concentration" are not fully independent. A recent study attributed the observed pattern of intensification of the dry season to human-induced climate change (mainly corresponding to increasing CO2) (Padrón et al., 2020).

Padrón, R.S., Gudmundsson, L., Decharme, B. et al. Observed changes in dry-season water availability attributed to human-induced climate change. Nat. Geosci. 13, 477–481 (2020). https://doi.org/10.1038/s41561-020-0594-1

2. Section 2.2. I would argue against a focus on statistical significance when presenting the results (see Amrhein et al., 2019). See specific suggestion in comment 7. Also, please clarify the maximum allowed complexity of the regression models when doing the stepwise multiple linear regression; are interaction terms included? Which are "all the possible subset regression models"? I would encourage the authors to include an example of the dominance analysis for a specific grid cell either in section 3.3 or in the supplement.

[Figure]

Amrhein, V., Greenland, S., and McShane, B. Scientists rise up against statistical significance. Nature. 567, 305–307 (2019). https://doi.org/10.1038/d41586-019-00857-9

3. Section 3.1 seems a bit short. Here it could be useful to mention possible confounding effects, for example, the positive correlation in Brazil between CO2 and NHD in Fig. 3b, could result from a higher NHD driven by changes in land cover (deforestation), which also coincide with the increasing trend in CO2.

4. In Fig. 4d clarify which sub-component of TWS is used. If it is a different sub-component for every grid cell it is perhaps useful to have a map in the supplement.

5. Lines 133–135. Expand or provide more evidence against the potential confounding effect of both soil moisture decreasing and NHD increasing as a result of higher incoming radiation (clear sky days). It may not necessarily be the case in all identified regions that lower soil moisture is exacerbating an increase in NHD. An additional analysis of the correlation between soil moisture and the evaporative fraction (i.e. latent heat / net radiation, these variables are also likely available in the employed reanalysis product) could clarify if there is indeed an effect of soil moisture limitation resulting on a higher partitioning towards sensible heat flux and therefore increased NHD.

6. Fig. 5 could benefit from also including the R2 for CO2 concentration.

7. It would be better to show the actual correlation values in Fig. 6. A two-panel figure with one map for shallow and one for deep soil depth could be a good option. Hatching could indicate which case has higher correlation.

8. Interpretation of Fig. 7 may also benefit from having two panels. One indicating only the total explanatory power of the joint influence, and another differentiating the regions where either factor is deemed more important. I understand this is a matter of personal taste.

9. Lines 173–176. It is likely related to the fact that low-middle-income regions are in the tropics, which are generally not water limited, and therefore tend to be more

sensitive to increased CO2. I do not find it necessary to have Fig. 8 in the main text.

---

## Author Comment (AC3) · 27 Jan 2021

**Authors' response to RC3**

**Black text: Referee comment**
**Blue text: Authors' response**

We greatly appreciate your recommendations for improving our manuscript. Our response and revision plans are provided below.

1. I think the representativeness of the statistical analysis should be placed in a better perspective and with clearer justifications. I am particular concern on how they attribute the correlations to the specific dominance of a driver. This requires a much in-depth elaboration throughout all the paper.

We admit that the description of statistical analysis in our previous manuscript is not clear enough, especially the dominance analysis method (Azen and Budescu (2003) (https://doi.org/10.1037/1082-989X.8.2.129)) that we applied to compare the relative importance of predictors in multiple regression. We will provide more detail of dominance analysis in a revised manuscript as follows:

"Stepwise multiple linear regression (Draper and Smith, 1998; Clow, 2010) is used to determine the significant (5% significance level assessed by an F-test) predictor variable in explaining NHD temporal variability for each grid cell. Next, the dominance analysis approach (Budescu, 1993; Azen and Budescu, 2003) is applied to compare the relative importance of those selected variables. The total variance among a set of predictors can be fully partitioned by dominance analysis even if the predictors are correlated (Vize et al., 2019). The issue of collinearity among predictors is addressed by examining the unique variance accounted for by the predictor across all possible regression sub-models involving the predictor. Dominance analysis is completed through an exhaustive set of pairwise comparisons among the predictors. The comparisons can be examined by three types of dominance: complete dominance, conditional dominance, and general dominance (Nimon & Oswald, 2013). To be completely dominant, a predictor must account for a greater amount of outcome variance than another predictor across every sub-model comparison. The conditional dominance of different predictors is conditional on what sub-model level is being examined. We applied the general dominance in this study, which is determined by taking the average amount of variance accounted for by a predictor across all sub-models and comparing it to other predictors. General dominance weights can be calculated for each predictor in a set and represent the relative proportion of $R^2$ attributable to a predictor."

2. I believe it is interesting to provide a global perspective, but I think the authors are in a position to provide more evidence at the regional scale. For instance, in Figure 3, they could select representative regions characterized by differentiated ecosystems and explain in more detailed the differences. At figure 3b, it is clear that the Amazonian basin has a contrast behavior in two areas that is also observed in Figure 7. Why? Could they please elaborate and provide more detail explanation supported by figures?

Thanks for pointing out such an interesting perspective that is worthy of further discussion. The Amazonian basin shows contrast behavior in two areas, which might be related to their different topography. In the part of the Amazonian basin where moisture deficit dominates elevated $CO_2$

in influencing the occurrence of hot extremes, topography is relatively flat. While in and to the south of the basin, topography is featured with mountain ranges and plateaus, where it is found elevated $CO_2$ is more important (Figure 7). In addition to this example, we will also discuss other representative regions characterized by different topographic and climatic patterns, in line with what the reviewer suggested based on ecosystems:

"In mountain ranges, for example, the Andes in South America, the Rockies in North America, and plateau sections such as the Brazilian Plateau and the Mongolian Plateau, interannual variability of root zone moisture is likely small due to steep topography, low energy input and temperature. In extreme dry regions, such as deserts in West Asia and North Africa (Sahara), root zone moisture is likely too small in volume to make significant impact on land surface energy partitioning. In those regions elevated $CO_2$ dominates moisture deficit in influencing the occurrence of hot extremes.

Areas where moisture deficit dominates elevated $CO_2$ in influencing the occurrence of hot extremes tend to be flat with thick soils, such as the North American Great Plain, the West Siberian plain and the North China Plain. Some moisture deficit dominant areas including India, Australia, South Africa, and the eastern tip of Brazil have one common characteristic that their interannual rainfall variability is high (Fatichi et al., 2012)." (see the figure shown below, Fig. 1 in https://doi.org/10.1175/JCLI-D-11-00356.1).

[Figure]

FIG. 1. The global map of $C_v$. Only stations with >50 yr of observations are included ($n = 8197$).

3. Closely connected to this, I miss more in-depth explanations on the causality of their effects. There are some attempts to explain a connection of feedbacks (lines 131-133), but in the majority of the results the reader is left out. In my opinion, at section 3.3 (I will call it discussion) the authors have an unique opportunity to provide some diagrams that show the feedback relations and the effects of the enhancement of CO2 or soil deficit in the hottest month.

Thanks for your suggestion, we will improve the discussion on causal relationship between hot extremes versus atmospheric $CO_2$ concentration and soil moisture by using a diagram with more detailed descriptions in the revised manuscript. As shown in the left panel of the diagram below, when infrared radiation is emitted by Earth's surface, some is absorbed by greenhouse gas and re-emitted in all directions by the atmosphere. Consequently, increasing GHG warms the Earth's surface and the lower atmosphere. As the major component of GHG, elevated $CO_2$ tends

to increase the likelihood of hot day occurrence.

The physical connection between soil moisture and hot extremes has been explained in the article *Extreme heat rooted in dry soils*, (Alexander, 2011, doi:10.1038/ngeo1045). Local soil moisture conditions strongly influence partitioning of net radiation on the surface into sensible heat and latent heat (right panel in the diagram shown below). Since the heat source of near-surface air during daytime is directly from sensible heat flux, it is reasonable to expect an association between number of hot days and soil moisture. The negative correlation can reflect a causal relationship between hot extremes and soil moisture deficit, either way, but in the hottest month it is more likely to reflect the feedback of dry soil to the lower atmosphere. Based on the mechanism that low soil moisture availability reduces evaporative cooling and increases atmospheric heating from sensible heat flux (Alexander, 2011), Mueller and Seneviratne (2012) (doi: 10.1073/pnas.1204330109) used correlation between hot days in the hottest month and 3-month SPI (a proxy for soil moisture) as coupling diagnostic to identify hot spots at a global scale. Those hot spots agree well with transitional climate regions (Koster et al., 2004 (doi: 10.1126/science.1100217); Seneviratne et al, 2010 (doi: 10.1016/j.earscirev.2010.02.004)) where soil moisture strongly constrains evapotranspiration variability and thus result in feedbacks to the atmosphere.

[Figure]

Although we have discussed the causal relationship between hot extremes and soil moisture deficit in the previously submitted manuscript, both reviewers mentioned that it was not clear enough. We will improve the relevant explanation in next revision.

4. In my opinion, there is a driver that is missing in the discussion: the water vapour pressure deficit. What is the role played by this variable in enhancing the warming of the hot extremes? Could they calculate it also using their wavelet analysis and then relate it to their findings?

Yes, both VPD and soil moisture have effects on surface energy partitioning and thus near-surface temperature. We did not include VPD in this analysis because the effect of VPD is strongly nonlinear, and on the other hand, reduced soil moisture (most likely occur in warm season (hottest months)) is reported to weaken the influence of VPD on surface energy partitioning (doi:10.1029/2006JD007161). In addition, Liu at al., (2020) (doi:10.1038/s41467-020-18631-1) compared the relative role of soil moisture and VPD in limiting ecosystem production at the global scale and reported that dryness stress on ecosystem production is

dominated by soil moisture. Therefore, we believe that the application of soil moisture alone can satisfy our study purpose.

We agree with you that discussion relevant to VPD is missing in our manuscript, we will modify it in the next revision.

5. The wording throughout the article is casual and not very exact. Would it be possible to identify the important ecosystems like the tropical rain forest, temperate or boreal forests instead of mentioning the continents (Africa, South America,...)?

We would like to explain that we describe the specific regions by continents follows the way that many previous studies used to identify hot spots of land-atmosphere coupling (e. g., doi/10.1175/JHM510.1; doi/10.1016/j.earscirev.2010.02.004; doi/10.1029/2010GL042764). We are pleased to adopt your suggestion to discuss representative regions characterized by different topographic and climatic patterns in the revised manuscript. The details are provided in our reply to your comment 2 above.

6. Figure 8 at section 3.3 appears out of the blue. In my opinion, it needs to be removed or much better embedded. As an alternative, un my opinion preferable, the extra space should be addressed to a more in-depth explanation of the cause-effects dominance of either enhanced CO2 or soil moisture deficit on the hot extremes.

We will follow your suggestion to remove Figure 8. In the next revision, we will add more explanation on the causality of effects of elevated $CO_2$ and soil moisture deficit on hot extremes, such explanation has been shown in our reply to your comment 3 above.

---

## Author Comment (AC4) · 4 Feb 2021

**Authors' response to RC4**

**Black text: Referee comment**

**Blue text: Authors' response**

The manuscript analyses the relative importance of global atmospheric $CO_2$ concentration and local soil moisture for the number of hot days of the hottest month (NHD). Much of the manuscript is focused on the analysis of the relationship between decomposed GRACE TWS and NHD. I agree that this is an interesting and novel aspect at the core of the manuscript, which should potentially be in the title (e. g. Relevance of TWS deficit and increased atmospheric CO2 for hot extremes).

Overall, the manuscript is well-structured and clear. Nevertheless, it would be useful to further clarify and improve some aspects of the methodology and presentation of the results. It doesn't seem like the best approach to focus on a direct comparison of moisture deficit and increasing CO2, given that changes in moisture deficit are also influenced by increasing CO2. Also, the computed correlations are taking into account both the trend and interannual variability. Looking separately at the trend and at the detrended interannual variability could be a good option. My hypothesis is that the trend in increasing CO2 correlates better with the trend in NHD, whereas the detrended interannual variability in NHD is clearly more related to the detrended interannual variability in local moisture deficit (the detrended variability in CO2 is likely very small). This can provide helpful context to better understand statements such as the conclusion in lines 180-182. One should not interpret these results as increasing CO2 is not too important for hot extremes; the fact that the correlation is not very high is likely because the detrended variability in CO2 is likely rather small and the study period relatively short.

We greatly appreciate your constructive comments for improving our manuscript, and your suggestion of an alternative title. Physically, it is root-accessible moisture that influences near surface air temperature. The wavelet decomposed TWS (raw TWS includes surface water, soil moisture, groundwater, ice, and snow) is applied here as one proxy of soil moisture. From this point of view, we believe the original title is more appropriate.

We agree with you that the impact of $CO_2$ concentration is more likely on the trend of NHD, while the soil moisture is on the interannual variability of detrended NHD. Dominance analysis we adopted in this study is able to quantify their relative importance in influencing the occurrence of hot extremes, even if the two predictor variables are correlated (detailed in our reply to your comment 2).

To address your concern that changes in moisture deficit might be influenced by increasing $CO_2$, we have examined the correlation between TWS and $CO_2$ in the figure shown below. Result shows that the two variables are mostly independent as there are very few grid cells (marked in black) having $p$ values no larger than 0.05.

Our analysis is based on (reconstructed) TWS, thus it only covers a short period (1985-2015). We agree with you that it is very likely that over a longer period, the relative importance of $CO_2$ may increase against soil moisture deficit. This can be examined when more TWS data become available in the near future. To make this message clear, we stated in the manuscript that "We note that under continuing increase of greenhouse gas forcing, hot extremes are expected to be

dominated by increased $CO_2$ concentration over larger areas in the future. Hence, global measures for reducing emissions are essential in combating current and future expansion of hot extremes."

[Figure]

Correlation coefficient ($r$) between raw TWS and $CO_2$. $p$ values of grid cells marked in black are smaller than 0.05.

**Specific comments**

1. In the first paragraph of the introduction, it could also be noted that "local moisture deficit" and "increased atmospheric CO2 concentration" are not fully independent. A recent study attributed the observed pattern of intensification of the dry season to human-induced climate change (mainly corresponding to increasing CO2) (Padrón et al., 2020).

Padrón, R.S., Gudmundsson, L., Decharme, B. et al. Observed changes in dry-season water availability attributed to human-induced climate change. Nat. Geosci. 13, 477–481 (2020). https://doi.org/10.1038/s41561-020-0594-1

This is an interesting and relevant study to our manuscript. It is not surprising that the two variables have some correlation. This is why we adopted the dominance analysis in our study. Following your earlier comment, we did look at the correlation of the two variables over the study period (1985-2015). The TWS and $CO_2$ is shown mostly independent. This seems to be inconsistent with what is shown in Padron et al (2020). Here are a few possible reasons to explain this apparent inconsistency: (1) Padron et al., (2020) identified dry-season water availability as annual minimum monthly P-ET where the dry-season is not always consistent with the hottest month identified in our study; (2) P-ET is not the same as soil moisture; (3) Padron et al., (2020) is based on data in 1902-2014 while our study focuses on the period of 1985-2015.

We will include Padron et al 2020 in the introduction section so that readers are made aware of this relevant research.

2. Section 2.2. I would argue against a focus on statistical significance when presenting the results (see Amrhein et al., 2019). See specific suggestion in comment 7. Also, please clarify the maximum allowed complexity of the regression models when doing the stepwise multiple linear regression; are interaction terms included? Which are "all the possible subset regression models"? I would encourage the authors to include an example of the dominance analysis for a specific grid cell either in section 3.3 or in the supplement.

Amrhein, V., Greenland, S., and McShane, B. Scientists rise up against statistical significance. Nature. 567, 305–307 (2019). https://doi.org/10.1038/d41586-019-00857-9

Thanks for recommending the useful reference. We agree to redraw Fig. 6 by using correlation coefficients instead of $p$ values. The new figure is shown in the reply to your comment 7.

There are 6 predictor variables (D1, D2, D3, D4, A4 and $CO_2$) included in the stepwise regression for each grid cell. All wavelet decomposed TWS series (D1-D4 & A4) are independent. TWS and $CO_2$ are also mostly independent as we replied above. In fact, even if those input variables are not independent, the dominance analysis can distinguish their relative importance as described in Azen and Budescu (2003) (https://doi.org/10.1037/1082-989X.8.2.129). We admit that the description of statistical analysis method especially the dominance analysis in the original manuscript is not clear enough. We will improve it in the next revision as follows:

"Stepwise multiple linear regression (Draper and Smith, 1998; Clow, 2010) is used to determine the significant (5% significance level assessed by an F-test) predictor variable in explaining NHD temporal variability for each grid cell. Next, the dominance analysis approach (Budescu, 1993; Azen and Budescu, 2003) is applied to compare the relative importance of those selected variables. The total variance among a set of predictors can be fully partitioned by dominance analysis even if the predictors are correlated (Vize et al., 2019). The issue of collinearity among predictors is addressed by examining the unique variance accounted for by the predictor across all possible regression sub-models involving the predictor. Dominance analysis is completed through an exhaustive set of pairwise comparisons among the predictors. The comparisons can be examined by three types of dominance: complete dominance, conditional dominance, and general dominance (Nimon & Oswald, 2013). To be completely dominant, a predictor must account for a greater amount of outcome variance than another predictor across every sub-model comparison. The conditional dominance of different predictors is conditional on what sub-model level is being examined. We applied the general dominance in this study, which is determined by taking the average amount of variance accounted for by a predictor across all sub-models and comparing it to other predictors. General dominance weights can be calculated for each predictor in a set and represent the relative proportion of $R^2$ attributable to a predictor."

As for your requirement to provide an example of dominance analysis for a specific grid cell, we would like to explain that we will provide more description of the method in the revised manuscript (the corresponding text is shown above). We hope that the new description of our analysis process is clear. On the other hand, the dominance analysis code is provided by the authors who proposed this method, which can be accessed from Azen and Budescu, (2003) (https://doi.org/10.1037/1082-989X.8.2.129).

3. Section 3.1 seems a bit short. Here it could be useful to mention possible confounding effects, for example, the positive correlation in Brazil between CO2 and NHD in Fig. 3b, could result from a higher NHD driven by changes in land cover (deforestation), which also coincide with the increasing trend in CO2.

We agree that more discussion on the results shown in Fig. 3 should be provided in Section 3.1. We will add the following discussion into the revised manuscript: "The effects of increased atmospheric $CO_2$ concentration on the occurrence of hot extremes are relevant for particular topographic and climatic conditions. For example, significant correlations are observed in extreme dry regions (e. g., the Sahara), mountain ranges (e. g., the Andes in South America), and plateau sections (e. g., the Mongolian Plateau and the Tibet Plateau)."

We will also mention possible confounding effects as you suggested: "Since deforestation can effect surface energy partitioning by reducing evapotranspiration, positive $CO_2$-NHD relationship in some areas of Southeast Asia and Brazil (Hansen et al., 2013) could result from the coincidence of deforestation-induced higher NHD and increasing $CO_2$ concentration." We appreciate that you brought this possible mechanism to our attention, which we didn't realize.

4. In Fig. 4d clarify which sub-component of TWS is used. If it is a different subcomponent for every grid cell it is perhaps useful to have a map in the supplement.

We will add a map showing which sub-component of TWS is used in Fig. 4d in the supplement as you suggested. The new figure is shown below:

[Figure]

Correlation between NHD and decomposed TWS components: (A) NHD-D1; (B) NHD-D2; (C) NHD-D3; (D) NHD-D4; (E) NHD-A4

5. Lines 133-135. Expand or provide more evidence against the potential confounding effect of both soil moisture decreasing and NHD increasing as a result of higher incoming radiation (clear sky days). It may not necessarily be the case in all identified regions that lower soil moisture is exacerbating an increase in NHD. An additional analysis of the correlation between soil moisture and the evaporative fraction (i.e. latent heat / net radiation, these variables are also likely available in the employed reanalysis product) could clarify if there is indeed an effect of soil moisture limitation resulting on a higher partitioning towards sensible heat flux and therefore increased NHD.

Thanks for providing the idea that analyzing the correlation between soil moisture and the evaporative fraction to clarify the effect of soil moisture on surface energy partitioning. We now completed the analysis as you suggested (see the figure shown below). Regions with relatively higher TWS-EF correlation are generally consistent with those identified moisture deficit dominant regions shown in Fig. 7, such as the US, South Africa, and Australia. We will add the additional analysis you suggested into the revised manuscript to improve the discussion on causal relationship between soil moisture deficit hot extremes.

[Figure]

Correlation between TWS and the evaporative fraction.

6. Fig. 5 could benefit from also including the R2 for CO2 concentration.

We agree to follow your suggestion to include $R^2$ for $CO_2$ in Fig. 5 in the revised manuscript. The new figure is shown below.

[Figure]

Histogram of the explanatory power (significant regression $R^2$) on NHD variability by using SPI, GLDAS_NOAH θ, raw TWS, decomposed TWS and atmospheric $CO_2$ concentration during 1985-2015.

7. It would be better to show the actual correlation values in Fig. 6. A two-panel figure with one map for shallow and one for deep soil depth could be a good option. Hatching could indicate which case has higher correlation.

Fig. 6 has been modified as you suggested. Correlation between NHD versus shallow (D1-D3) and deep (D4 and A4) soil moisture are shown in two panels. The corresponding description and discussion will be modified as: "Compared to D4+A4, D1-D3 shows higher correlation with NHD in parts of Asia and Europe, those regions are reported to have relatively shallower plant rooting depth (Fan et al., 2017). The central part of North America, the northeastern part of South America and the northwestern part of Southeast Asia are reported to have deeper plant rooting depths (Fan et al., 2017), where interannual variability (D4 and A4) of TWS seems to be more important than its seasonal variability (D1-D3) in explaining NHD temporal variability. This implies that plant water uptake from deeper soil plays an important role in θ-NHD coupling. However, D4+A4 also show stronger correlation than D1+D2+D3 with NHD in areas without deep roots, including the northern and southeastern parts of South America, parts of the Southeast Asia. This is because those regions have shallow groundwater table depth (Fan et al., 2013). It implies that in areas where groundwater is shallow, groundwater dependent ecosystems may contribute to heat mitigation, which is worthy of future investigation."

[Figure]

Correlation between NHD versus shallow (D1-D3) and deep (D4 and A4) soil moisture.

8. Interpretation of Fig. 7 may also benefit from having two panels. One indicating only the total explanatory power of the joint influence, and another differentiating the regions where either factor is deemed more important. I understand this is a matter of personal taste.

We have modified Fig. 7 as you suggested which is shown below. However, we think the original one seems to be more concise because all information can be shown in only one figure. We would like to explain that throughout the manuscript we use blue and red to represent moisture and $CO_2$, respectively. Therefore, in terms of color scheme, we believe the original Fig. 7 looks better. But if the reviewer and editor still suggest us to use the modified one, we are happy to do so.

[Figure]

(A) Spatial pattern of the total explanatory power of the joint influence of atmospheric CO2 concentration and soil moisture on hot extreme occurrences. (B) Spatial distribution of more important influencing factor for the occurrence of hot extremes between increased atmospheric CO2 concentration and local moisture deficit.

9. Lines 173-176. It is likely related to the fact that low-middle-income regions are in the tropics, which are generally not water limited, and therefore tend to be more sensitive to increased CO2. I do not find it necessary to have Fig. 8 in the main text.

We agree. Since we did not provide the mechanism for the relationship between income levels and the occurrence of hot extremes, we decide to remove Fig. 8 and the corresponding texts. This action is also consistent with what the Reviewer #2 suggested.